# Nowcasting of Surface Solar Irradiance Using FengYun-4 Satellite Observations over China

**Liwei Yang [1], Xiaoqing Gao [1,*], Zhenchao Li [1], Dongyu Jia [2] and Junxia Jiang [1,3]**

1   Key Laboratory of Land Surface Process and Climate Change in Cold and Arid Regions/Northwest Institute of Eco-Environment and Resources, Chinese Academy of Sciences, Lanzhou 730000, China
2   College of Geography and Environmental Engineering, Lanzhou City University, Lanzhou 730070, China
3   College of Earth and Planetary Sciences, University of Chinese Academy of Sciences, Beijing 100049, China
*   Correspondence: xqgao@lzb.ac.cn; Tel.: +86-0931-4967110

**Abstract:** The accurate prediction of surface solar irradiance is of great significance for the generation of photovoltaic power. Surface solar irradiance is affected by many random mutation factors, which means that there are great challenges faced in short-term prediction. In Northwest China, there are abundant solar energy resources and large desert areas, which have broad prospects for the development of photovoltaic (PV) systems. For the desert areas in Northwest China, where meteorological stations are scarce, satellite remote sensing data are extremely precious exploration data. In this paper, we present a model using FY-4A satellite images to forecast (up to 15–180 min ahead) global horizontal solar irradiance (GHI), at a 15 min temporal resolution in desert areas under different sky conditions, and compare it with the persistence model (SP). The spatial resolution of the FY-4A satellite images we used was 1 km × 1 km. Particle image velocimetry (PIV) was used to derive the cloud motion vector (CMV) field from the satellite cloud images. The accuracy of the forecast model was evaluated by the ground observed GHI data. The results showed that the normalized root mean square error (nRMSE) ranged from 18.9% to 21.6% and the normalized mean bias error (nMBE) ranged from 3.2% to 4.9% for time horizons from 15 to 180 min under all sky conditions. Compared with the SP model, the nRMSE value was reduced by about 6%, 8%, and 14% with the time horizons of 60, 120, and 180 min, respectively.

**Keywords:** satellite image; solar irradiance forecasting; FengYun-4A; surface solar irradiance; cloud motion

## 1. Introduction

Energy guarantees social progress and forms the basis for sustainable development. Under the pressure of environmental pollution and the shortage of traditional fossil energy, the world is now focusing its attention on the research of new energy sources. Solar energy is regarded as an important renewable energy source [1–3]. Therefore, photovoltaic power plants have developed rapidly in recent decades [4,5].

The outputs from photovoltaic power plants have the characteristics of great fluctuation and intermittence. Grid-connected large scale photovoltaic power plants create great difficulties in the management, dispatch, and safety operations of the power system. Accurate forecasting, regarding photovoltaic power, is extremely challenging in large-scale applications of photovoltaics (PV). The first step in predicting PV output is to forecast the surface solar irradiance as the fluctuation of surface solar irradiance, which is caused by the complex variation of clouds, affecting photovoltaic power output [5,6]. Therefore, the accurate temporal prediction of the surface solar irradiance is of great significance for the prediction, planning, and operational control of photovoltaic power.

Forecast methods of surface solar irradiance are classified according to different forecast horizons. For years and decades-ahead predictions, climate model simulations (e.g., CMIP5 model data) are used [7]. For days-ahead predictions, numerical weather prediction (NWP) models have mostly been used. For minutes and hours (0–4 h) ahead forecasts, ground-based cloud images and satellite data, combined with an artificial neural network (ANN) or statistical methods, are most commonly used [8–16].

In this paper, the forecasting method we studied belonged to very short-term (0–4 h) forecasting. For very short-term forecasting, many methods have been developed. M. Caldas and R. Alonso-Suárez [11] used all-sky images and irradiance measurements to forecast 1–10 min solar irradiance in advance near the city of Salto, Uruguay. Jane Oktavia Kamadinata et al. [12] used sky images and artificial neural networks (ANN) to forecast a 1–5 min global horizontal solar irradiance (GHI) in the Southeast Asia region. David Bernecker et al. [8] utilized sky images to forecast global horizontal solar irradiance from 5 seconds to 10 min in advance in Germany. Benali et al. [13] used three methods to predict three components of solar irradiation from the 1–6 h time horizons at the site of Odeillo, France: Smart persistence, artificial neural network, and random forest. Rosiek et al. [9] used satellite remote sensing data and an artificial neural network to forecast the building integrated photovoltaic (BIPV) power output with the horizon, up to 3 h ahead in Almería, Spain. Miller et al. [14] used a satellite-based model and fusion technique to forecast global horizontal solar irradiance with the horizons of 0–3 h in the USA. Arbizu-Barrena et al. [15] used advection and diffusion of the Meteosat Second Generation cloud index and weather prediction (NWP) model to forecast global horizontal solar irradiance (GHI) and direct normal irradiance (DNI), up to 6 h ahead in Spain. Wang et al. [16] used the Meteosat Second Generation (MSG) geostationary satellite to forecast global horizontal solar irradiance and direct normal irradiance with 0–4 h time horizons in the Netherlands.

The all-sky imaging measurement is often used for very short-term (1–15 min) solar irradiance forecasting. It has high spatial and temporal resolution, but the monitoring space is limited (within 5 km), and cloud movement is likely to be beyond detection for more than 1 h-ahead forecasts. The method of using artificial neural networks (ANN) to forecast solar irradiance relies on a large amounts of historical data, and the normalized root mean square error (nRMSE), is likely to be larger than in other methods that contains the cloud detection information. As satellite remote sensing observations have a wide vision and can better observe the variation of clouds, we focused on satellite observations to forecast the global horizontal solar irradiance in this paper.

In the application field of remote sensing satellites, the cloud index (CI), which calculates the atmospheric attenuation factor, has been successfully applied to the estimation of surface solar irradiance [17–20]. For example, the Heliosat 2 model has been widely used by many researchers to calculate the atmospheric attenuation factor [9,17–20]. The Heliosat model was developed for the Meteosat Generation satellite, is semi-empirical, and mainly applicable to Europe, which means that the model needs to be revised and calibrated when applied to other areas or satellites. As a result, many methods are only used for specific conditions. Therefore, a method to obtain the cloud index, developed for the FY-4A geostationary meteorological satellite, has not been found in the current literature [4–21]. Zhai et al. [22] developed a cloud/shadow detection method for remote sensing images that could detect the clouds with a relatively high accuracy and low computational complexity. Most of the multi/hyperspectral optical satellite sensors, with both visible and infrared spectral channels, have been tested with this method, and the results have proven that it can work well, although it has not estimated FY-4A remote sensing sensors (AGRI). The FY-4A satellite has a high spatial resolution (1 km × 1 km) in the visible and near-infrared (NIR) bands, so we used this method to calculate the clearness sky index derived from the cloud index.

In China, the area of arid and semi-arid regions account for about 47% of the total land area, and the arid area alone accounts for about a quarter of the land [23]. For the desert areas in Northwest China where meteorological stations are scarce, satellite observation data are very precious exploration data. Therefore, the model we used requires neither, historical data, nor the observed data of meteorological

factors, such as air temperature, humidity, and wind speed. Hence, it is suitable for the desert areas where the meteorological observation network system is not always available.

In this paper, an algorithm was developed to derive up to a 3 h forecast of global horizontal solar irradiance, using data from the AGRI sensor onboard FY-4A satellite, with the cloud index methodology (CSD-SI). The rest of this paper is organized as follows: Section 2 provides an overview of the FY-4A satellite, meteorological station, and clear sky model. Section 3 describes the cloud motion detection. Section 4 details the statistical index for the accuracy evaluation. Section 5 analyzes the forecast results in clear sky, partly cloudy sky, overcast sky, snowy sky, and all sky conditions. Section 6 summarizes the paper and looks toward future development.

## 2. Data Pre-Processing

In this section, we introduce the FY-4A remote sensing data and ground observation data, and then introduce the clear sky irradiance model.

### 2.1. FY-4A Satellite Data

The FY-4A satellite is a second generation of geostationary meteorological satellite in China. FY-4A was launched from the Xichang Satellite Launch Center on December 11, 2016. It was successfully located over the Equator at 99.5 degrees east longitude on December 17, 2016, and its current fixed point is over the Equator at 104.7 degrees east longitude. The AGRI sensor is one of the main loads of the FY-4A satellite; the remote sensing data in this work all came from the AGRI sensor. The AGRI sensor has 14 radiation imaging channels between 0.47 and 13.5 μm, covering six visible/near infrared bands, two medium infrared bands, two water vapor bands, and four long-wave infrared bands. The radiometric calibration accuracy and sensitivity of AGRI are 0.5, and 0.2 K, respectively. The one high spatial resolution is the visible channel (0.65 μm), which is 0.5 km × 0.5 km. In this work, the visible channel and the near infrared channel were used, where the spatial resolution is 1 km × 1 km. Figure 1 shows an example of an FY-4A satellite image, and it can be seen that the FY-4A satellite can accurately detect cloud information. The satellite observation data can be downloaded free of charge (http://data.cma.cn/).

### 2.2. Ground Data

Northwest China has rich solar energy resources and a large desert area, which has broad prospects for the development of PV plants. The meteorological station is located in Dunhuang (40°09′29.1″ N, 94°30′58.7″ E, 1148 m a.s.l.). The sensors used were Kipp and Zonen CNR4 net radiation sensors (Figure 2), in accordance with the International Organization for Standardization (ISO) standards. The instrument was regularly maintained (special weather and fixed time) and calibrated (every two years). The solar radiation data are measured by 10 second sampling and 2 min averaging. Furthermore, global solar radiation data from September 2018 to January 2019 are available.

The climatic condition of Dunhuang Gobi is a typical desert climate with low rainfall, large evaporation, and a large temperature difference between day and night. The annual sunshine duration is about 3246.7 h [24]. The underlying surface type is sandstone Gobi. Calculating the statistical average weather conditions (day and night) in Dunhuang over the past three years (2016–2018), we found that there were more cloudy and overcast days during the daytime, which were about 200 days in a year; and less rainy and snowy days of around 25 days in a year. Figure 2 shows the observation site at Dunhuang.

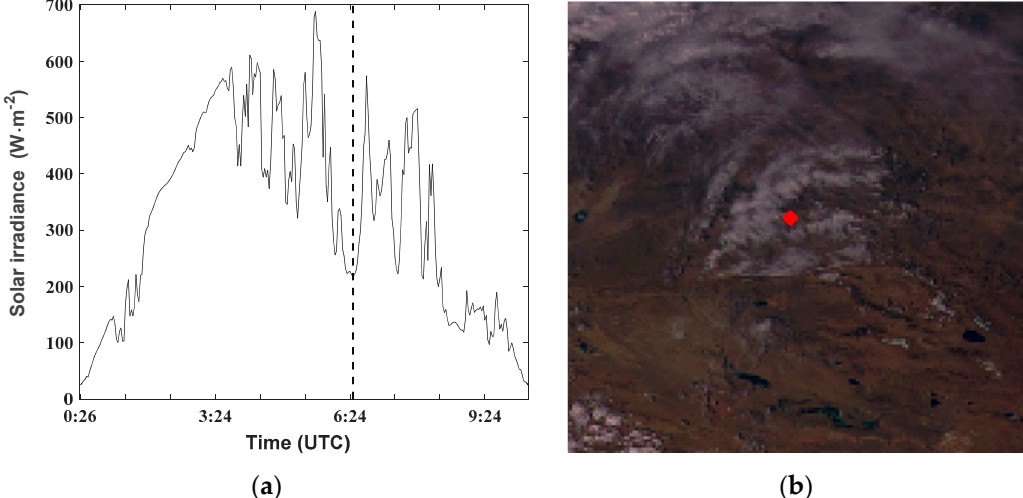

**Figure 1.** Example of (**a**) the diurnal variation of global horizontal solar irradiance (GHI) observed at the Dunhuang Gobi station on 1 November 2018 UTC and (**b**) the FY-4A satellite image on 1 November, 2018 at 06:30 UTC (the red point is the location of the Dunhuang Gobi station).

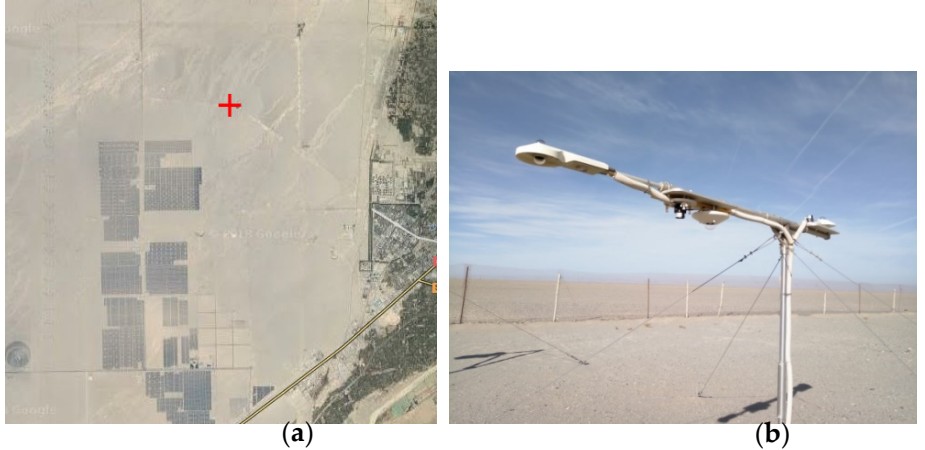

**Figure 2.** Photographs of the observation site at Dunhuang. (**a**) Google Map (the red cross indicates the location); (**b**) The instrument at the observation station.

### 2.3. Clear Sky Irradiance Model (ESRA)

To obtain the global horizontal solar irradiance value, we needed to calculate the solar irradiance in clear sky, and apply an index to estimate the atmospheric attenuation factor. The specific steps are as follows: The "atmospheric transparency coefficient" is the ratio of solar irradiance received on the Earth's surface to the solar irradiance received outside the atmosphere. This is denoted as $K_t$, and is written as Equation (1) as follows:

$$K_t = \frac{GHI}{G_0}. \tag{1}$$

where *GHI* is the global solar irradiance received on the Earth's surface and $G_0$ is the global solar irradiance received at the top of the atmosphere. Then, we can obtain the clearness sky index ($K_c$) by replacing $G_0$ with the clear sky solar irradiance ($GHI_{cs}$):

$$K_c = \frac{GHI}{GHI_{cs}}. \tag{2}$$

Next, we needed to calculate $GHI_{cs}$. In this work, the ESRA model, which is mainly designed to be used with satellite data, was used to calculate the clear sky global irradiance.

The global horizontal solar irradiance is the sum of direct irradiance and diffuse irradiance, so we first calculated these two variables. A more detailed introduction can be found in Rigollier et al. [25]. The direct radiation component ($I_{\mathrm{dir}}$) is given by,

$$I_{\mathrm{dir}} = I_0 \varepsilon \sin \alpha \exp(-0.8662 T_{\mathrm{L}} m \delta_{\mathrm{R}}(m)) \tag{3}$$

$$\varepsilon = 1 + 0.033 \times \cos(\frac{360 \times N}{365}) \tag{4}$$

where $I_0$ is the solar constant ($1367\ \mathrm{W \cdot m^{-2}}$ in this paper); $\varepsilon$ is the correction used for the variation in the sun–earth distance from its mean value; $\alpha$ is the solar altitude angle; $T_L$ is the Linke turbidity factor; $m$ is the air mass; and $\delta_{\mathrm{R}}(m)$ is the Rayleigh optical thickness. $N$ is the number of days from January 1 in a year (also known as Julian day).

The diffuse radiation component ($I_{\mathrm{dif}}$) is given by:

$$I_{\mathrm{dif}} = I_0 \varepsilon T_{\mathrm{rd}}(T_{\mathrm{L}}) F_{\mathrm{d}}(\alpha, T_{\mathrm{L}}). \tag{5}$$

The $T_{\mathrm{rd}}(T_{\mathrm{L}})$ and $F_{\mathrm{d}}(\alpha, T_{\mathrm{L}})$ are defined as follows,

$$T_{\mathrm{rd}}(T_{\mathrm{L}}) = -1.5843 \times 10^{-2} + 3.0543 \times 10^{-2} T_{\mathrm{L}} + 3.797 \times 10^{-4} T_{\mathrm{L}}^{2} \tag{6}$$

$$F_{\mathrm{d}}(\alpha, T_{\mathrm{L}}) = A_0 + A_1 \sin \alpha + A_2 [\sin \alpha]^2 \tag{7}$$

where $A_0$, $A_1$, and $A_2$ can be calculated from Equation (8):

$$\left\{ \begin{array}{l} A_0 = 2.6463 \times 10^{-1} - 6.1581 \times 10^{-2} T_{\mathrm{L}} + 3.1408 \times 10^{-3} T_{\mathrm{L}}^{2} \\ A_1 = 2.0402 + 1.8945 \times 10^{-2} T_{\mathrm{L}} - 1.1161 \times 10^{-2} T_{\mathrm{L}}^{2} \\ A_2 = -1.3025 + 3.9231 \times 10^{-2} T_{\mathrm{L}} + 8.5079 \times 10^{-3} T_{\mathrm{L}}^{2} \end{array} \right. \tag{8}$$

with the condition,

$$\text{if } (A_0 \cdot T_{\mathrm{rd}}) < 2 \times 10^{-3}, \text{then } A_0 = 2 \times 10^{-3} / T_{\mathrm{rd}}. \tag{9}$$

For the formulation of $T_{\mathrm{L}}$ considered here is [26]:

$$T_{\mathrm{L}} = 3.91 \tau_{550} \exp(0.689 * \frac{1013.25}{p}) + 0.376 \ln(u_{H_2O}) + [2 + 0.54 \frac{1013.25}{p} - 0.5(\frac{1013.25}{p})^2 + 0.16(\frac{1013.25}{p})^3] \tag{10}$$

where $\tau_{550}$ is the aerosol optical depth at 550 nm, $u_{H_2O}$ is the water vapor column value, $p$ is local atmospheric pressure.

Therefore, when the $I_{\mathrm{dir}}$ and $I_{\mathrm{dif}}$ values are obtained, the $GHI_{\mathrm{cs}}$ can be calculated from Equation (11):

$$GHI_{\mathrm{cs}} = I_{\mathrm{dir}} \cdot \cos \theta + I_{\mathrm{dif}} \tag{11}$$

where $\theta$ is the solar zenith angle.

Once $GHI_{\mathrm{cs}}$ has been obtained, the next step is to calculate the $K_{\mathrm{c}}$ value. The CSD-SI model, developed by Zhai et al. [22], requires the reflectivity of the visible and infrared spectral channels, which can be obtained from FY-4A satellite images, so this model was used to calculate the cloud index $n(0 < n < 1)$. Then, we get $K_{\mathrm{c}}$:

$$K_c = 1 - n. \tag{12}$$

Once $K_{\mathrm{c}}$ has been obtained, the next step was to calculate the wind vectors in pixels with the particle velocity method.

## 3. Methodology

In this section, we briefly introduce the persistence model, which mainly served as a reference model. Then, we introduce the cloud tracking method that was applied to the FY-4A satellite images. Next, we obtained the wind field in pixels, which can then be applied to the clearness sky index of the last satellite image. When the clearness sky index is forecasted, the global horizontal solar irradiance is forecasted by multiplying the clearness sky index with the clear sky solar irradiation. Figure 3 presents the flow chart of the main steps of global horizontal solar irradiance forecasting using the FY-4A satellite images, where the time horizons were from 15 to 180 min (for example, a 15 min time horizon represents a 15 min ahead forecasting). MATLAB 2016b software was used to model the prediction program.

### 3.1. Statistical Extrapolation Method

The persistence model assumes that the predicted value is the same as the previous one (Equation (13)). The smart persistence (SP) model was developed from the persistence model by Cyril Voyant [27] as Equation (14). The clear sky global horizontal solar irradiance model was used (Section 2.3) as it is the simplest method for solar irradiance forecasting. However, studies have shown that the stability of the prediction accuracy will decline with the increase in the forecasting time horizon [13].

$$\overset{\wedge}{GHI}_{t+\tau} = GHI_t \tag{13}$$

$$\overset{\wedge}{GHI}_{t+\tau} = GHI_t \bullet \frac{GHI_{cst+\tau}}{GHI_{cst}} \tag{14}$$

where the GHI with symbol "∧" represents the forecasted value and $\tau$ is the time step ahead at which the forecast is calculated from the given origin t. In our study, $\tau$ ranged from 15 to 180 min. This smart persistence model (Equation (14)) mainly served as a reference model in this work.

### 3.2. Cloud Tracking-Particle Image Velocimetry

Particle image velocimetry (PIV) is a method used to measure the velocity field of particles by recording the position of particles in multiple images and using the algorithm of image science [28–30]. Previous studies have shown that this method has a very high application value in the fields of fluid mechanics [15,28,29]. In this paper, we used the particle image velocimetry software "mpiv" [30] to derive the cloud motion vector (CMV) field from the FY-4A cloud images.

In order to ensure that the fastest moving clouds were within the scope of the research pixels, after many experiments, the pixel area of the satellite images was selected as $800 \times 800$ pixels, with the observation point in the center to reduce the consumption of computation resources. The "mpiv" method requires the selection of a square area of one cloud image to identify the most similar part (same size) in continuous images, in order to obtain the wind speed of this square area. In this work, the window size of the square area was set to 64 pixels and the overlap ratio of the adjacent window was set to 0. In order to obtain robust results, a filtering (global filter) and interpolation process (weighted averaging of adjacent data) was carried out and unrealistic changes were eliminated. Finally, the CMV field was derived. The velocity $\overrightarrow{V}$ (vector u and v pixel/min) passing through the observation station pixel was used to predict the future clearness sky index, where u and v refer to the velocities of the x and y coordinates, respectively. The shift parameter $\overrightarrow{a} = 1 + \overrightarrow{V} \cdot \Delta t$ was defined [29] and $\Delta t$ is the time horizon. With the shift parameter $\overrightarrow{a}$, we can obtain the clearness sky index value that will move to the observation station in the future horizons.

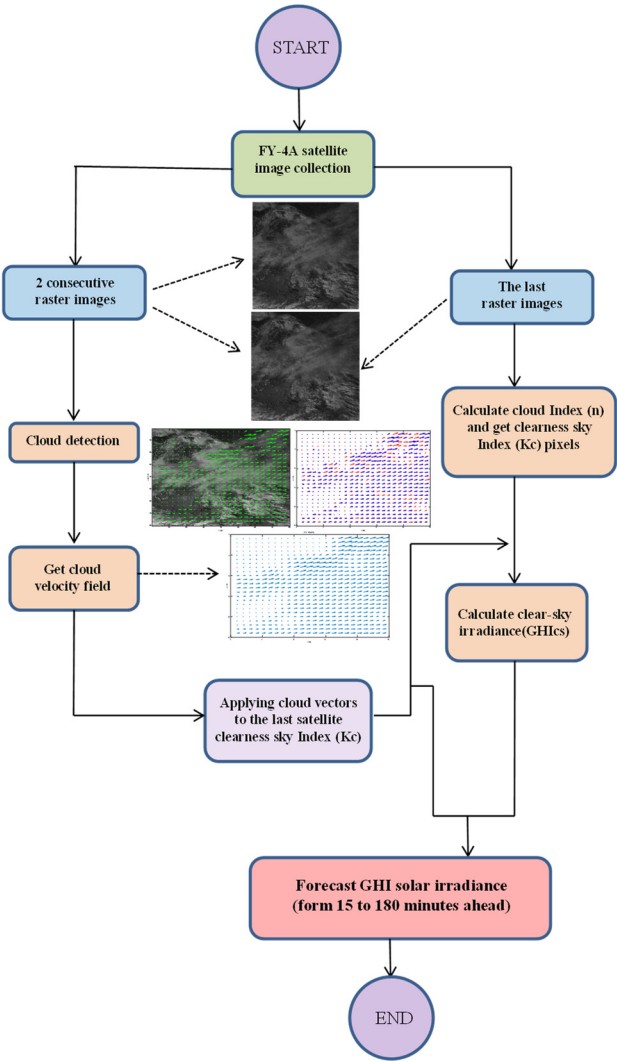

**Figure 3.** Flow chart of the main steps of global horizontal solar irradiance (GHI) forecasting with time horizons from 15 to 180 min.

## 4. Statistical Index for Accuracy Evaluation

In order to evaluate the accuracy of the model, statistical indexes are needed. According to previous studies [31–35], six statistical indexes were considered in this paper: The root mean square error (RMSE), the normalized root mean square error (nRMSE), the mean absolute error (MAE), the normalized mean absolute error (nMAE), the mean bias error (MBE), and the normalized mean bias error (nMBE).

The RMSE is the mean of the square root of the error between the forecasted value and the measured value. This index is very popular (Netflix machine learning competition evaluation method) [36], and is a quantitative weighting method. It is defined as,

$$RMSE = \sqrt{\frac{1}{N} \times \sum_{i=1}^{N} \left( X_{forecasted} - X_{measured} \right)^2} \tag{15}$$

where $X_{forecasted}$ is the forecasted value, $X_{measured}$ is the measured value, and $N$ is the number of observations.

The nRMSE is defined as:

$$nRMSE = \frac{RMSE}{\frac{1}{N}\sum_{i=1}^{N} X_{measured}} \times 100\%. \tag{16}$$

The MAE can avoid the problem of offset between deviations and describe the degree of data dispersion. It can be expressed by the equation

$$MAE = \frac{1}{N} \times \sum_{i=1}^{N} \left| X_{forecasted} - X_{measured} \right|. \tag{17}$$

The nMAE is defined as:

$$nMAE = \frac{MAE}{\frac{1}{N}\sum_{i=1}^{N} X_{measured}} \times 100\%. \tag{18}$$

The MBE is not as common when compared with other error metrics. The MBE is very similar to the MAE, except that absolute values are not used. It can determine whether the deviations are positive or negative in the model and is defined as:

$$MBE = \frac{1}{N} \times \sum_{i=1}^{N} \left( X_{forecasted} - X_{measured} \right). \tag{19}$$

The nMBE is defined as:

$$nMBE = \frac{MBE}{\frac{1}{N}\sum_{i=1}^{N} X_{measured}} \times 100\%. \tag{20}$$

## 5. Results and Discussion

In this section, we analyze the forecast results of global solar irradiance from 15–180 min time horizons under different sky conditions: clear sky, partly cloudy sky, overcast-sky, snowy-sky, and all sky conditions. The accuracy of the results is compared with the SP model.

### 5.1. Results under Clear Sky Conditions

Referring to the diurnal type index in [37], we classified the data according to weather conditions: Clear sky days, partly cloudy days, overcast days and rainy, and snowy days. For clear sky days, clouds do not appear from sunrise to sunset. We used the clear sky model above-mentioned (ESRA model) to calculate the surface solar radiation under clear sky conditions, where in this model, $T_L$ is a function of the aerosol optical (AOD) and the atmospheric water vapor [26,38], we used the Equation (10) to calculate $T_L$. The aerosol optical depth in the formula was selected as 0.1 [39] and the water vapor column value was selected as 0.09 cm in autumn and winter in Dunhuang Gobi [40,41]. Of course, there were some small deviations in both the model itself and the value selection of variables. Finally $T_L$ was selected as 2. The clear sky GHI can be estimated with any time horizons with a given location.

The accuracy of the model was validated by the observed data. Figure 4 shows the forecast results of the selected three days (108 samples) where the solar altitude angle was greater than 10°; it can be seen that the forecast values were very close to the measured values. Furthermore, Table 1 presents the statistical indexes for the clear sky model. The RMSE was about 14.48 W/m$^2$ and the nRMSE was 3.62%. The nMAE was about 2.51%, and the nMBE was about 1.07%. This means that the clear sky model had a high accuracy and could estimate the surface solar irradiance well under clear sky conditions.

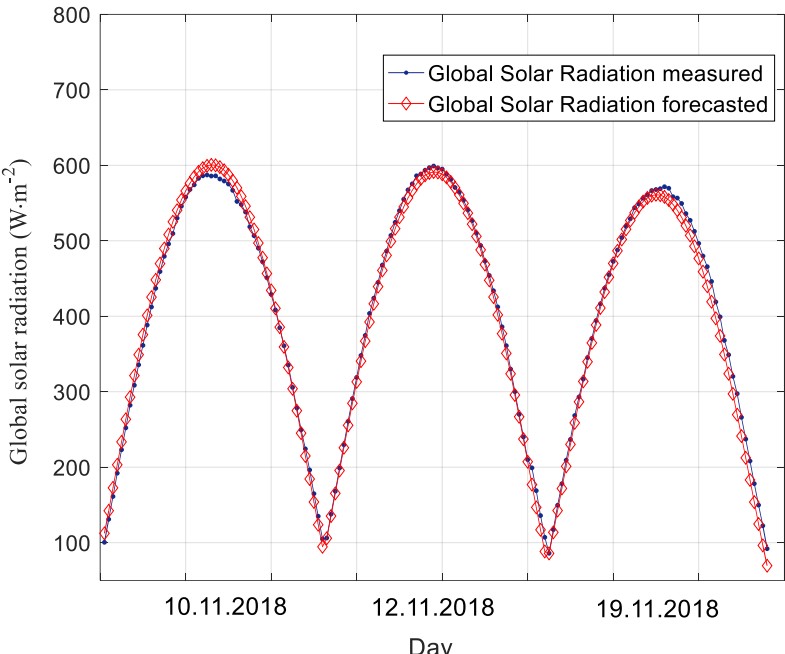

**Figure 4.** The forecast results of GHI under clear sky conditions (10.11.2018, 12.11.2018, 19.11.2018) for a total of 108 samples of the test set.

**Table 1.** Statistical indexes for the clear sky model (in W/m$^2$ for RMSE, MAE, and MBE).

|      | RMSE  | nRMSE (%) | MAE   | nMAE (%) | MBE  | nMBE (%) |
|------|-------|-----------|-------|----------|------|----------|
| GHI  | 14.48 | 3.62      | 10.02 | 2.51     | 4.27 | 1.07     |

### 5.2. Results under Partly Cloudy Conditions

Partly cloudy skies present sudden decreases and increases in solar irradiance, compared with sunny days, which are mainly due to the sudden appearance and disappearance of clouds at the location. Figure 5 shows the results of the forecasted global solar irradiance at a 15 min temporal resolution, under partly cloudy days, with a 60 min horizon for the selected three days (108 samples). The forecast time horizons were set to 15, 30, 45, 60, 120, and 180 min (the resolution of the forecast time horizon was set to 60 min after the one hour ahead forecasting, since the deviation almost presented a slowly increasing trend.). Therefore, Figure 5 is one example (the time horizon was 60 min). It can be seen from Figure 5 that the transient variation of clouds occurred on these days. The model could predict the basic trend of surface solar irradiance well, but there were some deviations between the forecasted and measured values during the whole day.

Table 2 presents the statistical indexes for the selected three days under partly cloudy skies with horizons from 15–180 min that were compared with the persistence model. Analyzing the table, for the model that we used the FY-4A satellite images, the RMSE value ranged from 79.0 W/m$^2$ to 85.1 W/m$^2$ from the 15-180 min horizons, the nRMSE value ranged from 21.1% to 24.9%, the nMAE value varied from 17.3% to 18.5%, and the nMBE value varied from 1.4% to 8.9%. The global solar irradiance values were overestimated (positive MBE and nMBE). For the persistence model (SP), it had a higher accuracy with the 15 min time horizon, where the RMSE value was about 73.3 W/m$^2$ and the nRMSE value was about 20.3%. However, with the increase of the time scale horizon, the RMSE increased to 135.2 W/m$^2$ and the nRMSE increased to 38.5% with the 180 min time horizon. Therefore, the forecasted results of GHI, under partly cloudy skies using FY-4A satellite images, were relatively stable with time horizons from 15–180 min. Compared with the SP model, the nRMSE was reduced by 5%, 10%, and 16% with the time horizons of 60 min, 120 min, and 180 min, respectively.

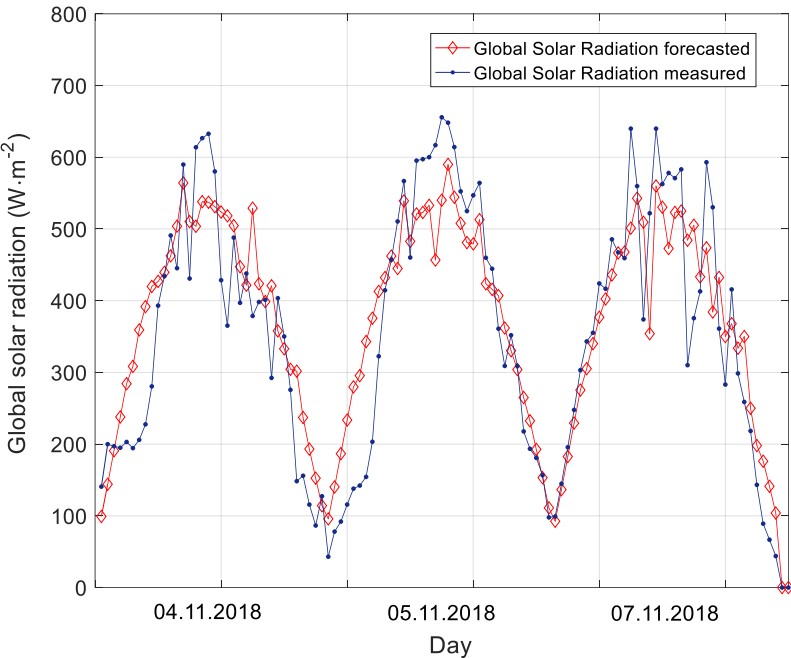

**Figure 5.** The forecasted results of GHI under partly cloudy skies with a 60 min horizon (04.11.2018, 05.11.2018, and 07.11.2018), total of 108 samples of the test set.

**Table 2.** Statistical indexes for partly cloudy skies with horizons from 15–180 min (in W/m$^2$ for the the root mean square error (RMSE), the mean absolute error (MAE), the mean bias error (MBE)).

| Metric | Model | t + 15 min | t + 30 min | t + 45 min | t + 60 min | t + 120 min | t + 180 min |
|---|---|---|---|---|---|---|---|
| RMSE | SP | 73.27 | 90.4 | 97.8 | 102.61 | 126.53 | 135.23 |
| | FY-4A | 79.02 | 82.62 | 79.89 | 82.23 | 85.09 | 82.18 |
| nRMSE (%) | SP | 20.3 | 25.03 | 27.08 | 28.14 | 35.04 | 38.45 |
| | FY-4A | 21.08 | 22.17 | 21.76 | 23.01 | 24.89 | 22.76 |
| MAE | SP | 51.43 | 65.9 | 72.31 | 76.51 | 95.95 | 97.28 |
| | FY-4A | 66.47 | 68.29 | 65.36 | 65.62 | 70.01 | 63.1 |
| nMAE (%) | SP | 14.24 | 18.25 | 20.03 | 21.19 | 26.57 | 27.12 |
| | FY-4A | 17.3 | 18.33 | 17.8 | 17.73 | 18.5 | 17.48 |
| MBE | SP | 2.03 | 4.45 | 6.84 | 9.05 | 16.38 | 25.03 |
| | FY-4A | 5.12 | 7.61 | 12.15 | 12.11 | 23.54 | 32.24 |
| nMBE (%) | SP | 0.56 | 1.23 | 1.89 | 2.51 | 4.54 | 6.89 |
| | FY-4A | 1.36 | 2.04 | 3.31 | 3.27 | 6.32 | 8.93 |

*5.3. Results under Overcast Conditions*

For overcast conditions, surface solar irradiance does not reach its maximum value in diurnal variation, and the variation is mainly caused by the persistent presence of clouds [42,43]. Figure 6 presents the forecasted results for the selected three days (108 samples) of global solar irradiance under overcast days with a 60 min horizon. It can be seen from the figure that the maximum value of GHI was around 550 W/m$^2$, and the deviation between the forecasted value and measured value at many points were large, more than 100 W/m$^2$. The forecasted values varied smoothly, while the measured values varied quickly.

Table 3 presents the statistical indexes for the selected three days under overcast skies with horizons from 15–180 min, and the SP model served as the reference model. For overcast days, the prediction accuracy was lower than the partly cloudy days. The RMSE value ranged from 93.8 to 116.8 W/m$^2$ from the 15–180 min horizons, the nRMSE value ranged from 31.1% to 38.8%, the nMAE value varied from 24.6% to 29.0%, and the nMBE value varied from 12.3% to 19.2%, so the GHI values were overestimated. For the SP model, the deviation was the minimum with the 15 min time horizons, the RMSE value was about 90.1 W/m$^2$, the nRMSE was about 29.9%, but the RMSE value increased

to 146.2 W/m$^2$, and the nRMSE increased to 48.5%, with the 180 min horizon. Compared with the SP model, the nRMSE values of the model we used were reduced by 8%, 6%, and 10% with the time horizons of 60 min, 120 min, and 180 min, respectively.

Thus, the prediction deviation of the two models both became larger when compared with the partly cloudy days. For overcast conditions, different layers and types of clouds may lead to complex atmospheres, which in turn, could lead to large errors in satellite inversion. For example, the influence of the cloud height on the GHI was very significant, and the atmospheric attenuation factor became very large by the presence of low clouds, due to the thick layer and high water content.

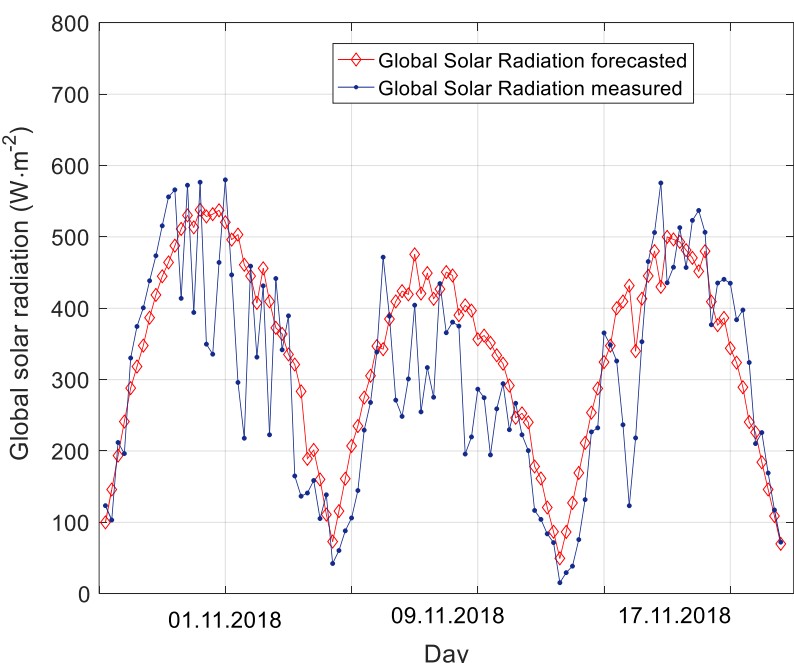

**Figure 6.** The forecasted results of GHI under overcast days with a 60 min horizon (01.11.2018, 09.11.2018, and 14.11.2018) for a total of 108 samples of the test set.

**Table 3.** Statistical indexes for overcast skies with horizons from 15–180 min (in W/m$^2$ for the RMSE, MAE, and MBE).

| Metric | Model | t + 15 min | t + 30 min | t + 45 min | t + 60 min | t + 120 min | t + 180 min |
|---|---|---|---|---|---|---|---|
| RMSE | SP | 90.05 | 103.51 | 124.07 | 117.83 | 123.42 | 146.15 |
| | FY-4A | 97.63 | 96.74 | 94.54 | 93.81 | 105.03 | 116.75 |
| nRMSE (%) | SP | 29.90 | 34.36 | 41.19 | 39.12 | 41.24 | 48.54 |
| | FY-4A | 32.41 | 32.11 | 31.39 | 31.14 | 34.87 | 38.78 |
| MAE | SP | 67.08 | 74.40 | 92.75 | 86.44 | 90.12 | 104.11 |
| | FY-4A | 78.11 | 76.74 | 74.10 | 74.29 | 79.54 | 87.46 |
| nMAE (%) | SP | 22.27 | 24.70 | 30.80 | 28.70 | 29.91 | 34.56 |
| | FY-4A | 25.93 | 25.48 | 24.61 | 24.67 | 26.41 | 29.03 |
| MBE | SP | 0.20 | 1.34 | 2.71 | 4.61 | 14.63 | 30.66 |
| | FY-4A | 40.01 | 39.19 | 38.45 | 37.00 | 47.09 | 57.86 |
| nMBE (%) | SP | 0.07 | 0.45 | 0.90 | 1.52 | 4.86 | 10.18 |
| | FY-4A | 13.30 | 13.01 | 12.76 | 12.28 | 15.63 | 19.21 |

*5.4. Results under Snowy-Sky Conditions*

For snowy days, the surface solar irradiance may fluctuate greatly and have no regularity. Sometimes, heavy and thick clouds have a very low clearness sky index ($K_c$), which greatly reduces the solar radiation on the Earth's surface, but the ground reflection increases due to snow cover, which in turn increases the GHI, therefore making it difficult to forecast [44,45]. We chose three days with snowfall, and Figure 7 shows the forecasted results of global solar irradiance under snowy days with

a 60 min horizon. The diurnal variation characteristics of surface solar irradiance were different, which may be due to the different thicknesses and types of clouds on snowy days. The deviations between the forecasted and measured values at many points were large; therefore, the prediction results of this model for the snowy days were poor.

Table 4 presents the statistical indexes for the selected three days under snowy skies with horizons from 15–180 min and compared with the SP model. For snowy days, the prediction accuracy was worst, the RMSE value ranged from 108.0 W/m$^2$ to 120.8 W/m$^2$, and the nRMSE value ranged from 43.8% to 49.2%, for the 15–180 min time horizons. For the SP model, the prediction accuracy was better than the model we used for the FY-4A satellite images from the time horizons of 15–60 min, and the nRMSE was reduced by about 5%. However, the prediction accuracy of the SP model declined sharply, and the nRMSE increased to 65.3% with the 180 min time horizon.

Therefore, in order to improve the accuracy of the snowy days forecast results, it is necessary to combine other products (cloud height and cloud thickness) in the future.

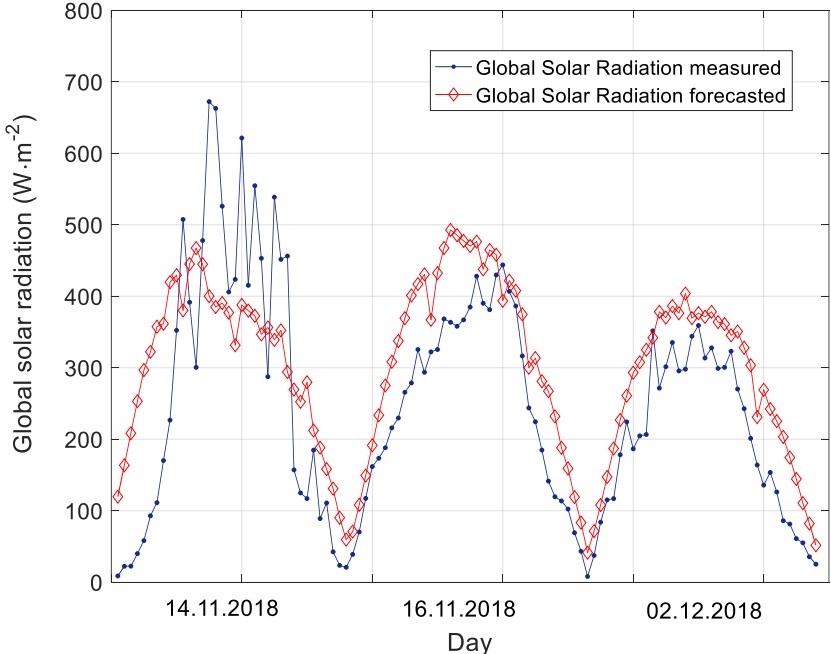

**Figure 7.** Forecasted results of GHI under snowy days with a 60 min horizon (14.11.2018, 16.11.2018, and 02.12.2018) for a total of 108 samples of the test set.

**Table 4.** Statistical indexes for snowy skies with horizons from 15–180 min (in W/m$^2$ for the RMSE, MAE, and MBE).

| Metric | Model | t + 15 min | t + 30 min | t + 45 min | t + 60 min | t + 120 min | t + 180 min |
|---|---|---|---|---|---|---|---|
| RMSE | SP | 75.69 | 90.03 | 99.48 | 109.23 | 140.51 | 156.48 |
| | FY-4A | 108.02 | 109.32 | 110.06 | 109.59 | 112.04 | 120.82 |
| nRMSE (%) | SP | 30.12 | 37.12 | 40.01 | 42.92 | 56.89 | 65.33 |
| | FY-4A | 43.79 | 44.37 | 45.05 | 44.64 | 45.61 | 49.21 |
| MAE | SP | 47.01 | 57.51 | 63.63 | 89.12 | 100.01 | 111.02 |
| | FY-4A | 91.09 | 91.99 | 94.85 | 92.31 | 93.23 | 103.93 |
| nMAE (%) | SP | 19.14 | 23.02 | 26.82 | 35.76 | 40.58 | 45.02 |
| | FY-4A | 37.19 | 37.39 | 38.84 | 37.60 | 38.02 | 42.33 |
| MBE | SP | 9.5 | 3.01 | 4.51 | 9.09 | 14.24 | 23.27 |
| | FY-4A | 55.13 | 54.12 | 56.61 | 54.42 | 66.15 | 83.34 |
| nMBE (%) | SP | 1.91 | 1.23 | 1.83 | 3.84 | 5.80 | 10.29 |
| | FY-4A | 21.87 | 22.05 | 23.06 | 22.17 | 27.01 | 33.95 |

## 5.5. Results under All Sky Conditions

In actual weather conditions, snowy days are still relatively rare at Dunhuang Gobi. We chose 10 consecutive days of data as the all sky condition samples, so a total of 360 samples (15.11.2018–24.11.2018). Figure 8 shows the forecasted results of global solar irradiance under all sky conditions with a 60 min horizon. It can be seen from Figure 8a that, although there were big deviations at some mutation points, the model could predict the basic trend of surface solar irradiance well. The good performance of the algorithm can be shown from the scatter plots and cumulative frequency curves in Figure 8b,c.

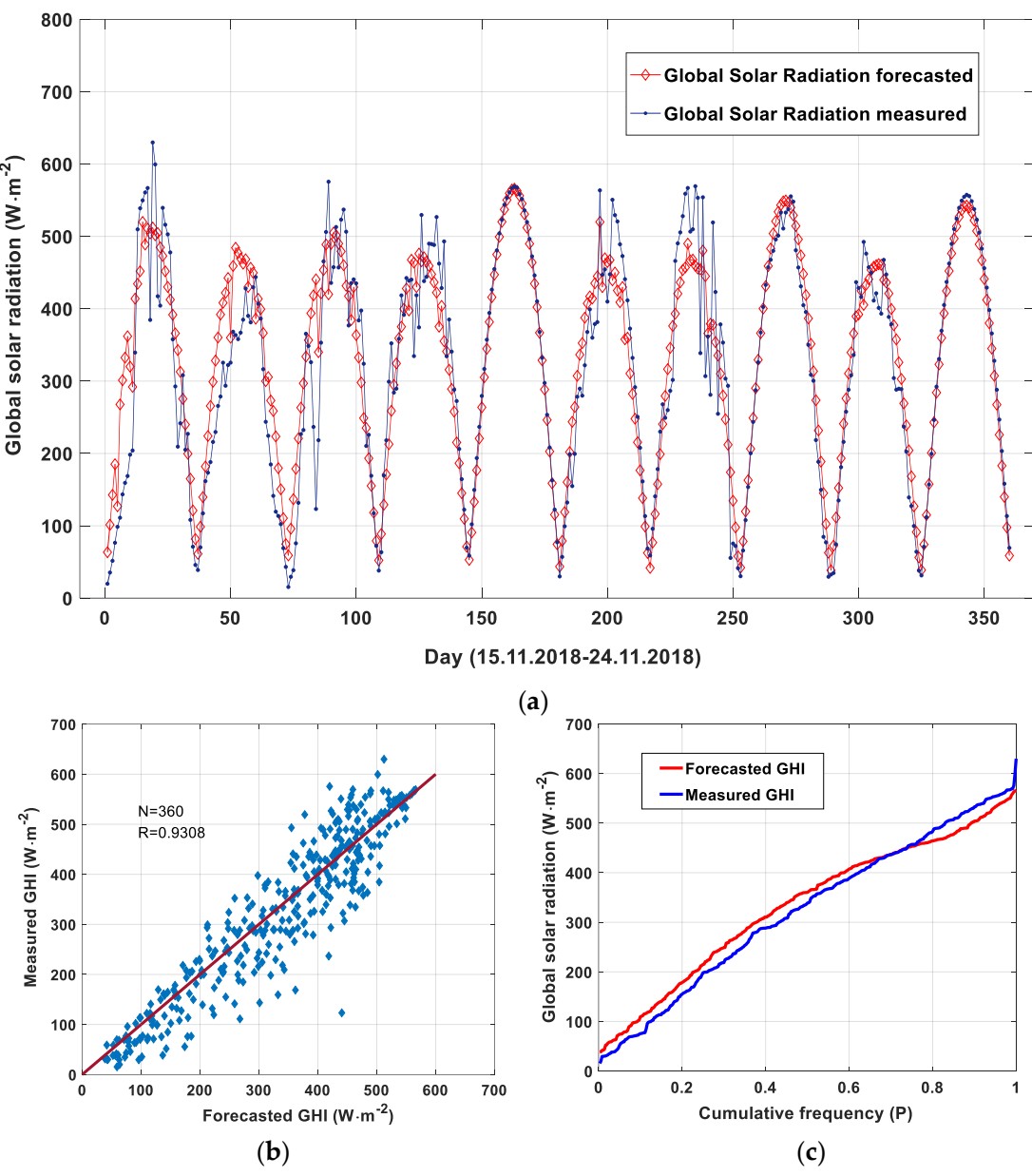

**Figure 8.** The forecasted results of GHI under all condition days with a 60 min horizon (15.11.2018–24.11.2018), total of 360 samples of the test set: (**a**) Continuous diurnal variation curves; (**b**) Scatter diagram; (**c**) Cumulative frequency curves.

Table 5 presents the statistical indexes under all skies condition with time horizons from 15–180 min, compared with the SP model. For all sky conditions, the RMSE value ranged from 60.9 W/m² to 68.6 W/m², the nRMSE value ranged from 18.9% to 21.6%, the nMAE value ranged from 13.9% to

15.9%, the nMBE value ranged from 3.2% to 4.9% for the 15–180 min horizons. The GHI values were overestimated (positive MBE and nMBE). For the SP model, it had a higher accuracy with the 15 min time horizon, where the RMSE value was about 54.6 W/m$^2$, and the nRMSE value was about 17.0%. However, with the increase in the time scale horizons, the RMSE value increased to 113.2 W/m$^2$ and the nRMSE value increased to 35.5% with the 180 min horizon. The prediction accuracy of the surface solar radiation using satellite cloud images was relatively stable with time horizons from 15–180 min, and the accuracy was relatively high, considering the current level of science and technology [46,47].

Figure 9 presents the nRMSE and nMAE values under different sky conditions, at different horizons. The results under a clear sky can be forecasted by any time horizon, so the value did not change with the time horizons. As seen in Figure 9, the nRMSE and nMAE values had similar trends with different horizons. The results under different sky conditions, ordered from highest to lowest in GHI prediction accuracy, were; clear sky, partly cloudy sky, overcast-sky, and snowy-sky conditions. The all sky conditions could best represent the actual weather conditions. For the SP model, the nRMSE and the nMAE value increased rapidly, with the increase in time horizons, and for the model using the FY-4A satellite cloud images, the nRMSE and the nMAE values showed no big changes in the 15–180 min time horizons.

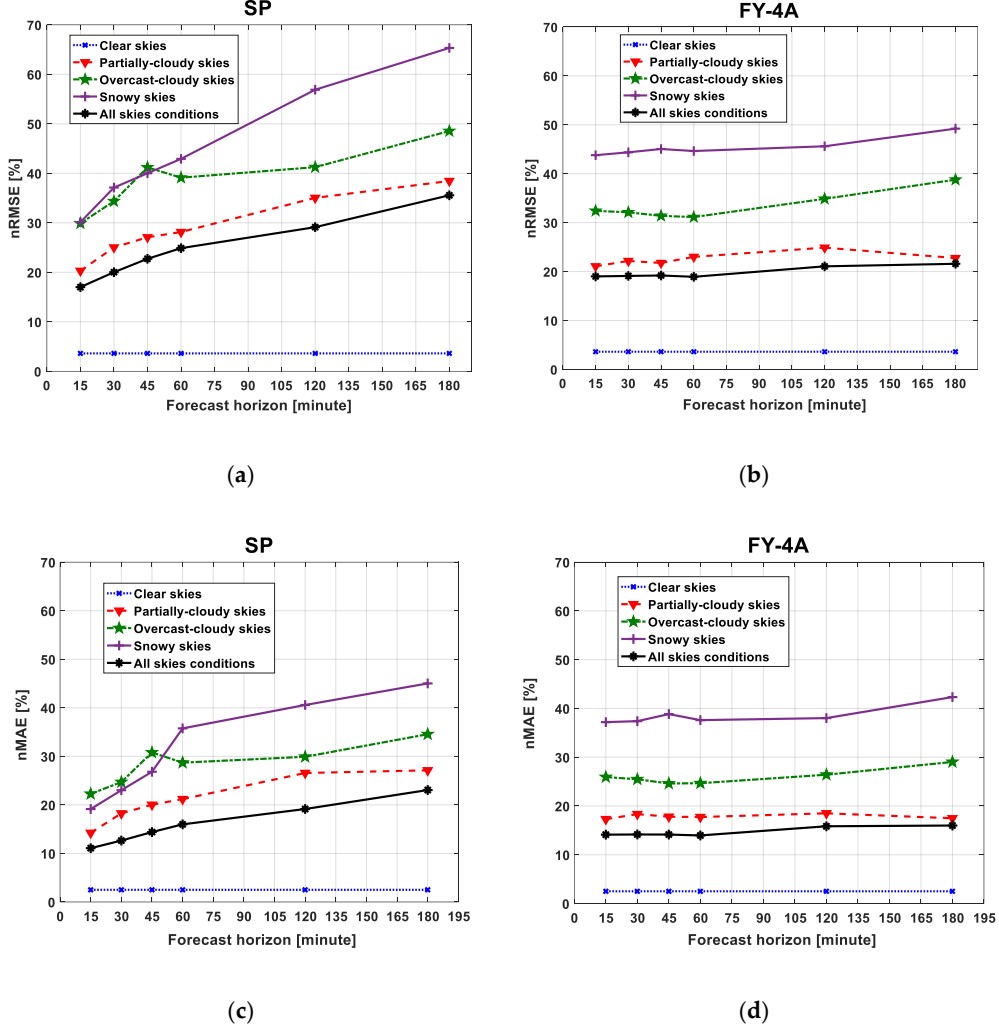

**Figure 9.** The nRMSE (%) and nMAE (%) under different sky conditions at different horizons: (**a**,**c**) SP model; (**b**,**d**) Model using the FY-4A satellite images.

**Table 5.** Statistical indexes under all sky conditions with horizons from 15–180 min (in W/m$^2$ for the RMSE, MAE, and MBE).

| Metric | Model | t + 15 min | t + 30 min | t + 45 min | t + 60 min | t + 120 min | t + 180 min |
|---|---|---|---|---|---|---|---|
| RMSE | SP | 54.58 | 63.95 | 72.58 | 79.43 | 92.99 | 113.54 |
| | FY-4A | 61.28 | 61.78 | 62.05 | 60.89 | 67.18 | 68.61 |
| nRMSE (%) | SP | 17.01 | 20.02 | 22.73 | 24.88 | 29.12 | 35.56 |
| | FY-4A | 19.01 | 19.11 | 19.20 | 18.93 | 21.07 | 21.59 |
| MAE | SP | 33.19 | 40.41 | 46.00 | 51.05 | 61.14 | 73.61 |
| | FY-4A | 45.26 | 45.76 | 45.72 | 44.87 | 49.93 | 50.21 |
| nMAE (%) | SP | 11.08 | 12.65 | 14.40 | 15.99 | 19.15 | 23.06 |
| | FY-4A | 14.12 | 14.16 | 14.14 | 13.96 | 15.84 | 15.99 |
| MBE | SP | −0.83 | −1.80 | −2.64 | −3.36 | −3.85 | −0.13 |
| | FY-4A | 11.39 | 11.50 | 11.58 | 9.84 | 14.79 | 14.83 |
| nMBE (%) | SP | −0.26 | −0.56 | −0.83 | −1.05 | −1.21 | −0.04 |
| | FY-4A | 3.39 | 3.56 | 3.58 | 3.18 | 4.74 | 4.87 |

## 6. Conclusions

Clouds have a great influence on the global solar irradiance received on the Earth's surface. The appearance, disappearance, and movement of clouds are very complex processes, which are affected by many factors. Therefore, the short-term forecasting of surface solar irradiance faces great challenges. In this work, a global solar irradiance forecasting method, using data from the AGRI sensor onboard the FY-4A satellite, with the cloud index methodology (CSD-SI) was presented. As the method does not require historical data, nor observational data of meteorological elements, the computational complexity of this method is low. Therefore, it is suitable for desert areas where the weather measurement system is not always available online.

To evaluate the accuracy of this method, the forecasted results of global solar irradiance under different sky conditions at different horizons from 15 to 180 min were presented and compared with the persistence (SP) model. For all sky conditions, the forecasted result of the nRMSE value varied from 18.9% to 21.6% and the nMBE value varied from 3.2% to 4.9% for time horizons from 15 to 180 min. The global solar irradiance values were overestimated. Compared with the SP model, the nRMSE value was reduced by 6% with the 60 min horizon, 8% with the 120 min horizon, and 14% with the 180 min horizon.

The best forecast results were presented for clear sky conditions, where the nRMSE value was approximately 3.6%, and the nMBE was about 1.1%. Therefore, the prediction results were very close to the actual measurement results, and the forecast results were very reliable under clear sky conditions.

For partly-cloudy conditions, the nRMSE value varied from 21.1% to 24.9% and the nMBE varied from 1.4% to 8.9% for time horizons of 15–180 min. Compared with the SP model, the nRMSE was reduced by 5%, 10%, and 16% with time horizons of 60 min, 120 min, and 180 min, respectively.

For overcast conditions, the nRMSE value varied from 31.1% to 38.8% for the time horizons from 15 to 180 min. Compared with the SP model, the nRMSE was reduced by 8%, 6%, and 10% with the time horizons of 60 min, 120 min, and 180 min, respectively. For snowy-sky conditions, the worst forecast results were presented with the nRMSE values, ranging from 43.8% to 49.2% for the time horizons from 15 to 180 min.

In conclusion, this work contributes to the field of solar irradiance prediction, using FY-4A satellite remote sensing data in China. The satellite remote sensing data are very precious exploration data in desert areas; however, these areas are often focused on the development of PV plants. The more proper observations of variables, related to the Linke turbidity factor for North West China, would help to improve the prediction capability of the algorithm. The current forecast algorithm performed well, but can be further improved in the future.

**Author Contributions:** X.G., L.Y. and Z.L. designed the research, L.Y. wrote the manuscript; Z.L. and L.Y. performed the field experiments; D.J. and J.J. contributed to the discussion.

**Funding:** This work is financially supported by the National Key Research and Development Program of China (2018YFB1502800), the National Natural Science Foundation of China (41805085), the National Natural Science Foundation of China (41675017), and the Opening Fund of Key Laboratory of Land Surface Process and Climate Change in Cold and Arid Regions, CAS (Grant LPCC2018006).

**Acknowledgments:** The authors would like to thank the National Meteorological Information Center for providing free access to the FY-4A satellite images. We would also like to thank M. K. Bah, J. Alonso-Montesinos, and G. Notton for their guidance. The authors would like to thank the Editors and Reviewers of the Remote Sensing journal for their valuable comments.

**Conflicts of Interest:** The authors declare no conflicts of interest.

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
