# Peer review of "Nowcasting of Surface Solar Irradiance Using FengYun-4 Satellite Observations over China"

_remotesensing, doi:10.3390/rs11171984_

Round 1
Reviewer 1 Report
This paper presents a study on the forecasting of GHI with new geostationary satellite, FengYun-4. The study is important for the solar energy researchers and the community in general since it is the first time a kind of studies like this is made in this region of the World. The paper is a well-written study, the structure of the article is excellent, however, the presentation of the methodology could be improved.
The paper is suitable for experts as well as appropriate for researchers working in related fields. There are some minor issues in the article that prevents its publication in the current form which with minor revision could be published in the Remote Sensing Journal.
Comments
1. how is the Linke Turbidity (TL) calculated?
2. how many days have been tested/validated in this experiment? It is recommendable to validate a one-year minimum.
3. Page 2. Line 44. numerical simulation models and NWP are the same based methods, same principles, explain better the differences
4. Page 2. Line 46. this is mostly days not months. this is called short-term forecast
5. Page 2. Line 47. Data, not dates.
6. More comments are included in the pdf

Reviewer 2 Report
The authors develop an algorithm to process FengYun-4 satellite image data to forecast surface solar irradiance in all-sky conditions, for photovoltaic energy systems.
As part of the introduction, it would help the reader to understand the reason behind accurate temporal prediction of the surface solar irradiance. What advantage does it give the power entity to have this information and what actions are required to be taken in anticipation of fluctuating solar irradiance?
In the clear-sky irradiance model of surface solar irradiance, there is no explanation for the form of the equations given, other than to refer to another paper. It was determined by the reviewer that the equations 3-10 come directly from the other paper, published in the journal Solar Energy. This reference paper outlines that the solar surface irradiance model was developed empirically, fitting ground-based measurements in Europe over a time period in the 1980's. The diffuse component of the model is based on a parameterization of atmospheric turbidity, TL. In the reference paper, Rigollier et al, 2000, the authors chose a specific value for TL, which should have been relevant for the European atmosphere during the time of the measurements. As TL is a function of the abundance or aerosols and gases such as water vapor, one would expect that the authors would address any differences in atmospheric turbidity between 1980's Europe and present day NorthWest China. There is no consideration of the applicability of the TL factor used where the authors quote the number '2', nor a discussion that the empirical model derived from European atmosphere relevant to the atmospheric / pollution conditions of the time period under which the model was derived, as compared to NorthWest China, which may be more arid, yet is subject to different aerosols. There is no discussion on why the number "2" is chosen.
It is not abundantly clear to this reviewer that this manuscript is appropriate for this journal, which I expect requires more through argument for an experiment's methodology at the outset.
Reviewer 3 Report
The authors have developed an algorithm for forecasting global solar irradiance from FY-4A satellite observations. The algorithm is intended for short-time scale forecasts and is applied to a desert region in China. The algorithm is then validated on the basis of in situ observations.
The methodology exploits well known concepts and algorithms already developed in literature. However, I have found the paper interesting because of its application to FY-AA. To my knowledge this is one of the first work which uses observations from FY-A4 and therefore I recommend the paper for publication.
However, my main concern is with English. Although I can understand what authors mean, the English is somewath clumsy and need to be improved. Also Eq. 12 and 13 could be re-written and notation simplified, e.g., I found t+min non appropriate, just use, t+Ď„ (the Greek letter "tau")
and say, "where Ď„ is the time step ahead at which the forecast is calculated from the given origin t. In our study Ď„ ranges from 15 to 180 min.
Reviewer 4 Report
@page { margin: 0.79in } p { margin-bottom: 0.1in; line-height: 115% }This work presents a method to derive up to 3h forecast of global horizontal irradiance using data from the AGRI sensor onboard FY-4A satellite. The method consists of calculating the cloud motion vectors via a particle image velocimetry (PIV), and the irradiance based on cloud index computation using other methods found in the literature.
The approach used by the authors is interesting and scientifically accepted.
However there are several major points to deal with, before the paper can be accepted for publication.
First of all I highly recommend an extensive revamping of the English language and style.
Moreover, some mistakes are reported here as an example:
24 - You either validate a model, or evaluate the accuracy
46 - do not start a sentence with “And” after a full stop. Please check through the manuscript.
47 – It should read “data” instead of “dates”
63,64 – amend as follows: … “(DNI) up to 6h ahead in Spain.”…
82: It should read “been” instead of “be”
96: remove “of the introduction”
110: “radiations” should be singular
149: correct the typo “radio” to “ratio”
208: I don’t think “transient” makes any sense here, please check.
279: It should read “partly-cloudy” instead of “partial-cloudy”, here and through the whole text
283-284: please be consistent using either min, mins, or minutes. I believe it should be min
307: use just “overcast condition”, without sky
320-322: Please be consistent with the tense all over the manuscript (e.g. the past is not right here)
337: Should it be high instead of low? please check
Besides, it’s hard to fully understand the meaning of the following statements:179-181, 308-309, 216-217, 284-285, 325-329. Please improve and explain further.
---Please emphasise the novelty element of this work with respect to the literature, both at the end of the Introduction, and in the Conclusion. I believe it lies in the combined use of the AGRI sensor with the cloud index methodology, is that correct? If there is more, please add.
---In order to compare two data series of the same variable, obtained with different methods (measured and simulated ones, for example), and to validate the methodology, the following estimators should be used for a complete validation: 1) scatter plot (measured and simulated values). 2) rmse, mbe, mae, correlation coefficient evaluation; 3) Comparison of the Cumulative frequency curves for measured and simulated values.
Some of these estimators are not evaluated and shown.
---325-329: This is interesting, I would have expected the other way round since overcast conditions are more steady than partly cloudy: could you please give some more insight about this.
--- Please explain further how you derive the cloud velocity field, how you extrapolate the motion and clarify the way to use the vector “a”; I would appreciate if you added a graphic example for better understanding. Besides, why (line 218) the overlap ratio is set to 0? Please also explain the filtering process, and what does it consists of.
--- In the flowchart, it appears you can go from start to end through only one branch: is this possible? I believe you need both branches to get to the end. Perhaps you should disconnect the last box “Forecast global…” from the “Applying cloud...” one? Please check, I might be wrong.
-264: I believe the TL acronym has not been introduced yet at this stage.
Figures:
- Suggestion: In all figures, you could remove “Global Solar Radiation” from the caption (since it is already clear from the y-axis label).
- The x-axis label can just read “Day”, instead of “Days of forecasted”.
- Fig.9: Please use different line styles in Figure 9 (dashed, solid, dot-dashed and so on) within each panel, otherwise it’s impossible to link the caption to the plot.
- Fig. 9: Please add a label on the top of the graphs, indicating which one refers to SP and to FY.
- Fig.9: Why is the time resolution no longer 15 min after the first hour of forecast? Please explain why in the main text.
References:
- Please check carefully the context of the citations, as some of these seem to be cited wrongly (e.g. the [21] does not use the Heliosat 2 model).
- Besides, the citations should list all names of the authors, please amend accordingly.
Round 2
Reviewer 1 Report
The paper is ok to be published
Author Response
Thank you very much for your recognition of our work.
Reviewer 2 Report
Dear Authors,
Thank you very much for your response to the question on how the Linke turbidity value = 2 was chosen. The explanation given in the response to this reviewer provides some insight. However, the method used needs to be outlined in the manuscript text itself, including the details of where various measurements are available. If I am finding these locations correctly, the SACOL station precipitable water vapor values are used for the site of the paper, Dunhuang Gobi, but SACOL station is in the middle of China, and ~1000 km from Dunhuang Gobi, so it is not clear that any PWV measurements from SACOL could be valid for the North West China site, and for the final Linke turbidity factor value.
“Variations of precipitable water in Northwest China During 1960-2015” and “Retrievals and analysis of precipitable water vapor and aerosol optical properties over semi-arid area of northwestern China” need to be cited with authors, and publication details. They are not searchable by the titles themselves.
The final selection of TL=2 in line 238 is terse, and the manuscript needs far more explanation regarding the measurements available, and applicability at the North West China site.
The Linke turbidity factor derived in your reviewer response comes to 2.3365. TL=2.3365 could entail a few percent difference in Direct Normal Incidence (DNI) as compared to TL=2 used in the manuscript, could it not? To evaluate the performance of this "nowcasting" method, error accounting should reflect the calculations for the range TL=2.3365 and TL=2, and discuss the errors given the uncertainty in the TL value. This discussion should also go into the conclusions, especially where clear-sky prediction errors are reported.
The conclusions should also mention that more proper determination of the Linke turbidity factor for North West China would help to improve the prediction capability of the algorithm.
Reviewer 4 Report
A remarkable update of the article has been done, but in my opinion it is still to do a check on the correspondence of the references. For instance Zhai et al. [22] is [23]. Please check them all.
Author Response
Thank you very much for your recognition of our work. We have checked the correspondence of the references. Revised portion are marked using the "Track Changes" function in Microsoft Word. Once again, thank you for the kind advice.
Yours sincerely,
Liwei Yang, Xiaoqing Gao, Zhenchao Li, Dongyu Jia, Junxia jiang